# Computed Tomography-Based Radiomics for Long-Term Prognostication of High-Risk Localized Prostate Cancer Patients Received Whole Pelvic Radiotherapy

**DOI:** 10.3390/jpm13121643

**Published:** 2023-11-24

**Authors:** Vincent W. S. Leung, Curtise K. C. Ng, Sai-Kit Lam, Po-Tsz Wong, Ka-Yan Ng, Cheuk-Hong Tam, Tsz-Ching Lee, Kin-Chun Chow, Yan-Kate Chow, Victor C. W. Tam, Shara W. Y. Lee, Fiona M. Y. Lim, Jackie Q. Wu, Jing Cai

**Affiliations:** 1Department of Health Technology and Informatics, Faculty of Health and Social Sciences, The Hong Kong Polytechnic University, Hong Kong SAR, China; abi_potsz@yahoo.com.hk (P.-T.W.); victorcw.tam@connect.polyu.hk (V.C.W.T.); shara.lee@polyu.edu.hk (S.W.Y.L.); jing.cai@polyu.edu.hk (J.C.); 2Curtin Medical School, Curtin University, GPO Box U1987, Perth, WA 6845, Australia; curtise.ng@curtin.edu.au; 3Curtin Health Innovation Research Institute (CHIRI), Faculty of Health Sciences, Curtin University, GPO Box U1987, Perth, WA 6845, Australia; 4Department of Biomedical Engineering, Faculty of Engineering, The Hong Kong Polytechnic University, Hong Kong SAR, China; saikit.lam@polyu.edu.hk; 5Department of Oncology, Princess Margaret Hospital, Hong Kong SAR, China; lmy084@ha.org.hk; 6Department of Radiation Oncology, Duke University Medical Center, Durham, NC 27708, USA; jackie.wu@duke.edu

**Keywords:** artificial intelligence, biomarker, machine learning, malignancy, medical imaging, prognosis, progression-free survival, radiation therapy, recurrence, tumor

## Abstract

Given the high death rate caused by high-risk prostate cancer (PCa) (>40%) and the reliability issues associated with traditional prognostic markers, the purpose of this study is to investigate planning computed tomography (pCT)-based radiomics for the long-term prognostication of high-risk localized PCa patients who received whole pelvic radiotherapy (WPRT). This is a retrospective study with methods based on best practice procedures for radiomics research. Sixty-four patients were selected and randomly assigned to training (*n* = 45) and testing (*n* = 19) cohorts for radiomics model development with five major steps: pCT image acquisition using a Philips Big Bore CT simulator; multiple manual segmentations of clinical target volume for the prostate (CTV_prostate_) on the pCT images; feature extraction from the CTV_prostate_ using PyRadiomics; feature selection for overfitting avoidance; and model development with three-fold cross-validation. The radiomics model and signature performances were evaluated based on the area under the receiver operating characteristic curve (AUC) as well as accuracy, sensitivity and specificity. This study’s results show that our pCT-based radiomics model was able to predict the six-year progression-free survival of the high-risk localized PCa patients who received the WPRT with highly consistent performances (mean AUC: 0.76 (training) and 0.71 (testing)). These are comparable to findings of other similar studies including those using magnetic resonance imaging (MRI)-based radiomics. The accuracy, sensitivity and specificity of our radiomics signature that consisted of two texture features were 0.778, 0.833 and 0.556 (training) and 0.842, 0.867 and 0.750 (testing), respectively. Since CT is more readily available than MRI and is the standard-of-care modality for PCa WPRT planning, pCT-based radiomics could be used as a routine non-invasive approach to the prognostic prediction of WPRT treatment outcomes in high-risk localized PCa.

## 1. Introduction

According to Global Cancer Statistics, prostate cancer (PCa) was the third most common cancer accounting for 7.3% of all cancer deaths in 2020 [1]. In 2023, the most common male cancer in USA was PCa causing an estimated 34,700 deaths, which is the second highest cancer death rate of 11% [2]. As per the European Society for Medical Oncology (ESMO) [3] and American Cancer Society (ACS) [4] guidelines, patients with localized prostate cancer can be classified into three main risk groups based on T category, Gleason score (GS) and prostate-specific antigen (PSA) representing low, intermediate and high risks, respectively: T1-T2a and GS ≤ 6 and PSA ≤ 10; T2b and/or GS 7 and/or PSA 10–20; and T3a or GS 8–10 or PSA > 20. More than one third of PCa patients belong to the high-risk group [5]. 

Low- and intermediate-risk patients may only need active surveillance. However, either long-term androgen deprivation therapy (ADT) plus radical radiotherapy (RT) or radical prostatectomy (RP) and pelvic lymphadenectomy are required for treating high-risk patients [3]. Whole pelvic RT (WPRT) and prostate-only RT (PORT) are the two typical radical RT options used for treating high-risk prostate cancer patients [6,7,8]. Usually, the Roach formula is used to estimate involvement of pelvic nodes based on GS and PSA, with 15% or greater nodal risk as an indicator for adopting WPRT despite its increased acute and late gastrointestinal toxicity compared to PORT [6,7,9]. Nonetheless, a recent literature review on the identification and prediction of prostate cancer indicated that PSA and GS may not be reliable prognostic markers. This is because PSA can increase without PCa, and intermediate- and high-risk patients may have low PSA levels. Also, the variation of GS determined from pre- and post-RP specimens is common [10]. It is noted that more than 40% of high-risk patients die from PCa, which is 10 times greater than for low-risk patients [5]. Hence, better approaches to PCa risk stratification, treatment selection and outcome assessment have been explored over the years, and radiomics is considered one of the potential candidates [10,11].

Radiomics refers to quantitative feature extraction from medical images as imaging biomarkers for clinical decision support with the aim of improving the accuracy of diagnosis, prognosis and outcome prediction, which are essential in personalized medicine (also known as precision medicine) and include diagnosis and treatment [10,12]. Although the concept of radiomics has only emerged over the last decade, numerous studies have explored its potential in precision medicine including for the prognostication of prostate cancer [10,11,12,13,14,15,16,17,18,19,20,21,22,23,24,25,26,27,28,29,30,31,32,33,34,35,36,37,38]. So far, the benefits of radiomics have not been translated into clinical practice because of its limited reproducibility as a result of a lack of process standardization [10,12]. Typically, five major steps are involved in the radiomics workflow including medical image acquisition and segmentation, feature extraction and selection, and model development [10,11,12,13,14]. However, the approaches involved in each step varied across studies in terms of different scanning protocols for image acquisition and the use of manual, semi-automatic or fully automatic segmentation. These have subsequent impacts on the reproducibility of results because features determined as clinically relevant to one setting become irrelevant to another setting when images are acquired and segmentation is performed in varying ways [10,11,12,17,18,19,20,21,22,23,24,25,26,27,28,29,30,31,32,33,34,35,36,37,38]. Commonly, magnetic resonance imaging (MRI) [17,18,19,20,21,22], positron emission tomography (PET) [23,24,25,26,27,28,29,30,31,32,33,34,35] and computed tomography (CT) [36,37,38] are used for PCa diagnosis and management [10]. However, CT is the standard-of-care modality for PCa RT planning, while the other modalities may not be available in some clinical settings [39]. Also, the use of MRI for radiomics appears problematic due to its non-standardized voxel intensity values, which are greatly influenced by the variation in scanning protocols [10,17,18,19,20,21,22]. Although planning CT (pCT)-based radiomics allows seamless integration into existing RT workflow, there is a paucity of studies on this for high-risk PCa. These include that published in 2019 on PCa risk stratification and our latest study published in 2023 on pCT-based radiomics for the long-term prognostication of high-risk localized PCa patients who received PORT [10,11,12,13,39]. To the best of our knowledge, no study has explored the potential of pCT-based radiomics with its counterpart, WPRT. Given the high death rate of high-risk PCa (>40%) and the reliability issues associated with traditional prognostic markers [5,10], the purpose of this study is to investigate pCT-based radiomics for the long-term prognostication of high-risk localized PCa patients who received WPRT. We hypothesized that pCT-based radiomics could be used as a routine non-invasive approach for the prognostic prediction of WPRT treatment outcomes in high-risk localized PCa. 

## 2. Materials and Methods

This is a retrospective study with methods based on Lambin et al.’s [12] best practice procedures for radiomics research derived from their radiomics quality score instrument. The best practice procedures employed in our radiomics workflow included multiple segmentations, feature reduction to avoid overfitting, cutoff analyses, use of discrimination statistics such as receiver operating characteristic curve (ROC) and area under the ROC curve (AUC), and a three-fold cross-validation resampling method [12,13]. This study was conducted in accordance with the Declaration of Helsinki, and approved by the Institutional Review Board of The Hong Kong Polytechnic University (approval number: HSEARS20200902001 and date of approval: 20 September 2020), and Clinical & Research Ethics Committee of New Territories East Cluster of Hospital Authority of Government of Hong Kong Special Administrative Region (approval number: NTEC-2020-0633 and date of approval: 9 December 2020). 

### 2.1. Patient Selection

Eighty-four high-risk localized PCa patients, who received treatments between May 2009 and October 2014, and met the following inclusion criteria were identified through the electronic health record system of Princess Margaret Hospital, Hong Kong Special Administrative Region. The inclusion criteria were as follows: those with risk of pelvic lymph node involvement estimated by the Roach formula ≥15%; and whose WPRT were received [7,9]. The identified patients were excluded for the following reasons: second malignancies other than PCa; previous PCa treatment; unavailability of pre-treatment biopsy results; or death unrelated to PCa. Eventually, sixty-four patients were selected and randomly assigned to training (*n* = 45) and testing (*n* = 19) cohorts for the radiomics model development. Their clinical and WPRT treatment data such as age, pre-treatment TNM stage, GS, PSA, WPRT technique, dose fractionation, ADT drug regimen, follow-up duration and clinical outcome and Digital Imaging and Communications in Medicine (DICOM) datasets (pCT images and structure sets) were collected accordingly [36,37,38]. Figure 1 summarizes the patient selection procedures.

### 2.2. ADT and WPRT Treatment

All selected patients were given neoadjuvant ADT (2 weeks of flutamide and 2 injections of 3-month luteinising hormone-releasing hormone agonist (LHRHa)) prior to WPRT. The WPRT’s clinical target volume (CTV) for the prostate (CTV_prostate_) was given 70–76 Gy in 2 Gy per fraction over 7–8 weeks with static field intensity-modulated radiotherapy or volumetric modulated arc therapy (VMAT). CTV for whole pelvic lymph nodes (CTV_LN_) was given 44 or 50 Gy with three-dimensional (3D) conformal radiotherapy or VMAT. All treatment plans were computed to meet acceptance criteria and organs at risk (OARs) constraints. Details on CTV, planning target volume (PTV), acceptance criteria and OARs constraints are given in Appendix A. After completion of WPRT, patients were prescribed adjuvant LHRHa for up to 3 years, and there were follow-ups at intervals of 3–6 months for disease monitoring. The PSA levels were determined and evaluated at each visit. Imaging tests were performed when an increase of PSA was found [6,7,9,40].

### 2.3. Clinical Endpoint

This study’s clinical endpoint was the six-year progression-free survival (PFS) of patients after WPRT. This referred to patients not having any distant metastasis, local recurrence, regional recurrence and/or chemical recurrence for six years after completing the WPRT course. Patient deaths unrelated to PCa were censored [9]. 

### 2.4. Radiomics Workflow

#### 2.4.1. Medical Image Acquisition

Non-contrast pCT scans were performed on all selected patients using the Koninklijke Philips N.V. Brilliance Big Bore CT simulator (Amsterdam, The Netherlands) as per in-house protocol. Patients were required to empty their bladders and then drink 400 cc of water an hour before the scans to achieve comparable bladder status. The images were taken with the patients in the treatment position (both hands on the chest in the supine position and the use of customized foam for immobilization) and the following scan parameters—tube voltage: 120 kV; tube current: 350–450 mAs; slice thickness: 1.5 or 3 mm; field of view: 60 cm; matrix size: 512 × 512; pixel spacing: 1.18; and a standard convolution kernel for image reconstruction [41].

#### 2.4.2. Medical Image Segmentation

All the collected DICOM structure sets including OARs (bladder, bowel, femoral head, penile bulb and rectum), CTV and PTV (prostate and lymph nodes) were manually contoured by a radiation oncologist experienced in prostate cancer radiotherapy using the ‘Draw Planar Contour’ function of the ‘Contouring’ interface on the Eclipse version 13 treatment planning system (Varian Medical Systems, Palo Alto, CA, USA). These were subsequently reviewed and approved by another radiation oncologist with associate consultant grade or above on the same system for original clinical use. To adhere to Lambin et al.’s [12] best practice procedure for segmentation, an additional consultant radiation oncologist was involved in reviewing and approving these DICOM structure sets, including the CTV_prostate_ as volume of interest (VOI), on the Eclipse version 13 treatment planning system based on the European Society for Therapeutic Radiology and Oncology (ESTRO) consensus guideline for this study [39,42]. The definitions of level of apex, lateral, anterior and posterior borders and the base of the prostate as well as seminal vesicles stated in the ESTRO consensus guideline were used to check the accuracy and consistency of the CTV_prostate_ manual segmentation to minimize intra- and inter-observer variabilities. For example, the level of prostate apex was defined as about 1 cm above the upper border of the penile bulb. Complete details on these definitions are available in the ESTRO consensus guideline [42]. The average number of consecutive pCT slices segmented for the CTV_prostate_ was 21 (standard deviation [SD]: 3 and range: 15–28). Figure 2 shows pCT image examples with the manually delineated CTV_prostate_ contours included in the study.

#### 2.4.3. Feature Extraction

The pCT image pre-processing and feature extraction procedures used in this study were based on those of the Image Biomarker Standardization Initiative (IBSI) [43], and were performed using the open-source Python-based radiomics feature extraction package, PyRadiomics version 2.2.0 [10,44]. Uniform volumetric spacing was achieved through isotropic resampling by resizing the images to 1 × 1 × 1 mm^3^ based on linear interpolation. Subsequently, a constant intensity resolution was attained by discretizing the images to a fixed bin width of 10 Hounsfield units (HU) to extract texture features. Also, the Laplacian of Gaussian (LoG) filter with 0.5, 2, 3, 4, 4.5 and 5 mm sigma values was used to reconstruct the images for feature extraction from various scales of edge detection and image smoothing. Shape features (*n* = 14), first-order features (*n* = 126) and texture features (*n* = 511) of the CTV_prostate_ were extracted after the pre-processing of images as per Appendix A. The shape, first-order and texture features described the 3D size and shape and voxel intensity distribution of the CTV_prostate_ and voxel intensity relationship within the CTV_prostate_ sub-regions, respectively. For every pCT dataset, 651 radiomic features were extracted in total [43].

#### 2.4.4. Feature Selection

Statistical approaches were used to consecutively select a smaller set of features for our model from the training data. The training cohort features were scaled and centered to avoid under- or over-presentation of individual ones. Additionally, the mean and SD of the scaled and centered training data were used to normalize the testing cohort features. A Mann–Whitney U test was conducted to determine the clinical association of every radiomic feature for its selection. Features having no statistically significant differences across the outcome groups (*p* > 0.05) were removed. Also, redundant features were identified based on the pair-wise correlation of the features using Spearman’s rank correlation coefficient. When the absolute correlation coefficient of two features was greater than or equal to 0.4, the feature with the greater mean absolute correlation was removed. The model was then developed based on all remaining features [12,39,45,46,47,48]. A correlation coefficient of 0.4 was selected as the threshold because previous radiomics studies used it to indicate moderate correlation with promising outcomes for feature reduction [49,50].

#### 2.4.5. Model Development

The model development was based on logistic regression with a least absolute shrinkage and selection operator (LASSO) penalty as well as three-fold cross-validation. The LASSO penalty was used for prediction error reduction and model simplification. It enabled the most predictive feature selection through the penalization of the sum of feature coefficient absolute values. Features that had minor contributions to the model were forced to undergo coefficient reduction to become zero and subsequently being removed. The three-fold cross-validation involved randomly dividing the training data into three groups. Two out of three groups were employed to train with the other reserved for validation. This process was repeated three times to involve each group once in the validation. In addition to the model testing and bias minimization, the three-fold cross-validation was also responsible for identifying the optimal regularization parameter for LASSO (lambda). Finally, 1000 models were developed as a result of repeating the process of model training a 1000 times [12,39,45,46,47,48].

#### 2.4.6. Statistical Analysis

The statistical analysis was performed using R version 3.6.3 (The R Foundation, Indianapolis, IN, USA). The R packages used include the following: base package for randomization and normalization; stats package for chi-squared test, Fisher’s exact test, Mann–Whitney U test and Spearman’s rank correlation coefficients; caret package for pair-wise correlations; glmnet package for logistic regression with the three-fold cross-validation and LASSO penalty; and ROCR and cvAUC packages for ROC analysis and AUC calculation. A *p*-value of less than 0.05 represented statistical significance [12,39,45,46,47,48]. 

Models with the lowest number of selected features were used for radiomics signature development. Every feature coefficient (*β*) and intercept within the radiomics signature was determined by taking the average of those values of the included models. Equation (1) illustrates the radiomics signature and was used to calculate the radiomics score (Rad-score) for every patient [51,52].
(1)$$Rad-score=\sum_{i=1}^{n}\beta_{i}\times {feature}_{i}+intercept$$

The cutoff of the Rad-score was determined based on the evaluation of model accuracy, sensitivity and specificity. The cutoff was used to classify whether a patient was more likely to have six-year PFS based on their Rad-score. The performance of the derived radiomics signature was evaluated in terms of accuracy, sensitivity and specificity. Additionally, the average AUC values of the training and testing cohorts were calculated [12,39,45,46,47,48]. Figure 3 summarizes the feature selection, model development and statistical analysis processes.

## 3. Results

Table 1 shows the clinicopathological characteristics of the included patients. There was no statistically significant difference found between the characteristics of the training and testing cohorts. Regarding the clinical endpoint, 81.5 months was the median PFS of all patients, and 80.0% and 78.9% of patients in the training and testing cohorts had six-year PFS, respectively. There were 13 included patients (20.3%) with metastasis and/or recurrence in six years after completing the WPRT course, constituting 20.0% of the training and 21.1% of the testing cohorts. 

Among the 1000 developed models, 799 models with the fewest (two) selected features were used for radiomics signature development. Both selected features were textural: run entropy of grey level run length matrix after LoG filtering with a sigma value of 2 mm (RE-GLRLM_σ2mm_); and small area emphasis of grey level size zone matrix after LoG filtering with a sigma value of 4.5 mm (SAE-GLSZM_σ4.5mm_). Both RE-GLRLM_σ2mm_ and SAE-GLSZM_σ4.5mm_ had statistically significant differences in the feature values between patients with and without six-year PFS (*p*-values: 0.0208 and 0.0191), respectively. The developed radiomics signature is illustrated in Equation (2).
Rad-score = 0.291 (RE-GLRLM_σ2mm_) + 0.358 (SAE-GLSZM_σ4.5mm_) − 1.47(2)

The average AUC values of the developed model for the training and testing cohorts were 0.756 (95% confidence interval (CI): 0.756–0.757) and 0.707 (95% CI: 0.706–0.707), respectively (Figure 4). With the cutoff determined as a third-quartile value (i.e., −1.11), patients were stratified into high- (Rad-score ≥ −1.11) and low- (Rad-score < −1.11) risk groups, which refer to unlikely and more likely to have six-year PFS, respectively (Figure 5). The respective accuracy, sensitivity and specificity of the radiomics signature were 0.778, 0.833 and 0.556 in the training cohort and 0.842, 0.867 and 0.750 in the testing cohort.

## 4. Discussion

In our study, the key radiomic features among the large arrays of data extracted from the CTV_prostate_ of the pre-treatment pCT images were selected to develop a two-feature radiomics signature to predict the six-year PFS in high-risk localized PCa patients with WPRT as the primary treatment. Highly consistent predictive performances were achieved by our model with average AUC values of 0.76 and 0.71 in the training and testing cohorts, respectively. The consistent performances could be attributed to the fact that the pCT for the external beam RT is highly standardized and calibrated for dose calculation, hence improving results reproducibility [10,11,12,17,18,19,20,21,22,23,24,25,26,27,28,29,30,31,32,33,34,35,36,37,38]. This potentially addresses one of the major issues associated with radiomics, which is the inability to translate benefits into clinical practice [10,12]. 

According to the review on the radiomics used for the identification and prediction of PCa published in 2021, most studies have focused on PET radiomics because PET is a functional imaging modality that provides detailed information on cell metabolism and proliferation, morphology, perfusion, receptor density and tumor viability, which are important for this identification and prediction task [10,23,24,25,26,27,28,29,30,31,32,33,34,35]. Although MRI might not be available in some settings, it is suggested that MRI should be a standard-of-care modality for PCa diagnosis [53,54]. Hence, there are more studies on MRI radiomics than CT for PCa identification and prediction [10,17,18,19,20,21,22,36,37,38]. However, a review on MRI radiomics for PCa risk stratification published in 2023 showed that only three studies used MRI to predict biochemical failure after receiving RT, with two reporting the AUC values of their models [55,56,57,58]. In Dinis Fernandes et al.’s study, their model achieved an AUC value of 0.63 [57]. Although Zhong et al.’s model was able to attain a mean AUC value of 0.99 during training, it reduced to 0.73 in testing [58]. This highlights one main limitation of the use of MRI in radiomics: its non-standardized voxel intensity values are greatly affected by scanning protocol variations, resulting in less reproducible results [10,17,18,19,20,21,22]. In contrast, our study’s model attained average training and testing AUC values of 0.76 and 0.71, demonstrating higher reproducibility despite the training AUC value of 0.71 being a little lower than that of Zhong et al.’s model at 0.73 [58]. Similarly reproducible model performance results were also shown in our previous study on CT radiomics for long-term prognostication of high-risk localized PCa patients who received PORT (mean training and testing AUC: 0.798 and 0.795, respectively) [39]. Additionally, CT is more readily available than MRI and PET and the standard-of-care modality for PCa RT planning, which allows for seamless integration into the existing RT workflow. These could be considered as other merits of CT radiomics for long-term prognostication of high-risk localized PCa patients who received RT [11,39,53]. 

Our developed radiomics signature with the determined cutoff of −1.11 again achieved consistent accuracy, sensitivity and specificity between training (0.778, 0.833 and 0.556) and testing (0.842, 0.867 and 0.750) cohorts to stratify patients into high- (Rad-score ≥ −1.11) and low- (Rad-score < −1.11) risk groups. Our radiomics signature consists of two texture features: GLRLM Run Entropy (RE-GLRLM_σ2mm_) and SAE-GLSZM Small Area Emphasis (GLSZM_σ4.5mm_). RE-GLRLM_σ2mm_ quantifies the heterogeneous texture pattern within the CTV_prostate_ by representing the variations in the allocations of run lengths and grey levels. SAE-GLSZM_σ4.5mm_ measures the quantities of smaller-sized zones and fine textures within the CTV_prostate_ by representing the distribution of consecutive voxels that share identical intensity values. As these two features have positive weightings in our radiomic signature with higher values of RE-GLRLM_σ2mm_ and SAE-GLSZM_σ4.5mm_, the Rad-score becomes greater and indicates the CTV_prostate_ of patients showing more heterogeneous 3D patterns. Also, a higher Rad-score represents a higher possibility of disease progression within six years after completing WPRT, which is in line with a previous study’s findings that a more heterogenous PCa tumor has greater resistance to therapies [59]. Similar investigations have been conducted on other malignancies showing a variety of texture features correlating with angiogenesis and hypoxia, which could be used to indicate the aggressiveness of breast cancer [60,61] and distant metastasis for nasopharyngeal carcinomas [62]. Hence, these show that radiomics is a viable approach to extract the distinctive characteristics of a malignant mass and quantify the respective heterogeneity to determine the prognosis and therapeutic response to oncological diseases [63].

Clinical failure or biochemical failure after primary RT is common in PCa patients. About 30–50% of patients are affected by biochemical failure within 10 years after RT [64]. Clinical failure occurs in approximately 25% of patients with biochemical failure within eight years with symptoms because of disease recurrency [65,66,67]. Palliative approaches, such as observation and ADT, are eventually employed to manage many of these patients [68,69,70]. However, curative intent salvage treatments—e.g., salvage prostatectomy, brachytherapy, stereotactic body radiotherapy, etc.—can be applied to selected patients with biochemical failure or isolated local recurrences without coexisting metastatic lesions [71,72,73]. Our radiomics model would be useful for the pretreatment identification of patients with a higher likelihood of disease progression after treatment, resulting in better clinical decision making and patient management, e.g., the use of state-of-the-art imaging examination to follow-up with these patients, increasing opportunities to offer salvage treatments to them when still applicable. In this way, personalized or precision medicine could be realized [10,12,74]. 

Our study has several limitations. It is a retrospective study with a relatively small sample size of 64 patients from one single center. According to Lambin et al.’s [12] radiomics quality score instrument, a prospective study with data collected from multiple sites would be a better design as this allows for model external validation [12,75,76]. However, our arrangement should be considered acceptable because some recent CT radiomics studies on identification and prediction of PCa also retrospectively collected patient datasets from one single site with comparable sample sizes of 69–80 patients [36,37,38]. Despite multiple manual segmentations of the CTV_prostate_ as per the ESTRO consensus guideline to address the potential intra- and inter-observer variability issues and the selection of ≥ 0.4 Spearman’s rank correlation coefficient for feature reduction based on previous radiomics studies, our model’s generalizability needs to be confirmed by assessing the intra- and inter-observer variability and the effect of other correlation coefficient threshold settings in future studies [12,42,49,50]. Nonetheless, this is the first study on pCT radiomics for the long-term prognostication of high-risk localized PCa patients who received WPRT, which could further justify our study design. Given the promising results of this study, future studies with a larger number of datasets collected prospectively from multiple centers with assessments on the intra- and inter-observer variability and the effect of various correlation coefficient settings is warranted for our model’s external validation and to confirm its generalizability. It is noted that deep learning (DL) has become popular in medical imaging [75,76,77,78,79,80]. Hence, another direction for further study is to develop a DL-based radiomics model for the long-term prognostication of high-risk localized PCa patients after WPRT [10].

## 5. Conclusions

This study’s results show that our pCT-based radiomics model was able to predict six-year PFS in high-risk localized PCa patients who received WPRT as the primary treatment with highly consistent performances (mean AUC: 0.76 (training) and 0.71 (testing)) and was comparable to other similar studies including those on MRI-based radiomics. The accuracy, sensitivity and specificity of our radiomics signature that consists of two texture features, namely GLRLM Run Entropy (RE-GLRLM_σ2mm_) and SAE-GLSZM Small Area Emphasis (GLSZM_σ4.5mm_), were 0.778, 0.833 and 0.556 (training) and 0.842, 0.867 and 0.750 (testing), respectively. Since CT is more readily available than MRI and PET and is the standard-of-care modality for PCa RT planning, pCT-based radiomics can be used as a routine non-invasive approach to the prognostic prediction of WPRT treatment outcomes in high-risk localized PCa. Nonetheless, further study on the external validation of our model is warranted to ensure that its benefits can be realized in clinical settings to achieve personalized or precision medicine.

## Figures and Tables

**Figure 1 jpm-13-01643-f001:**
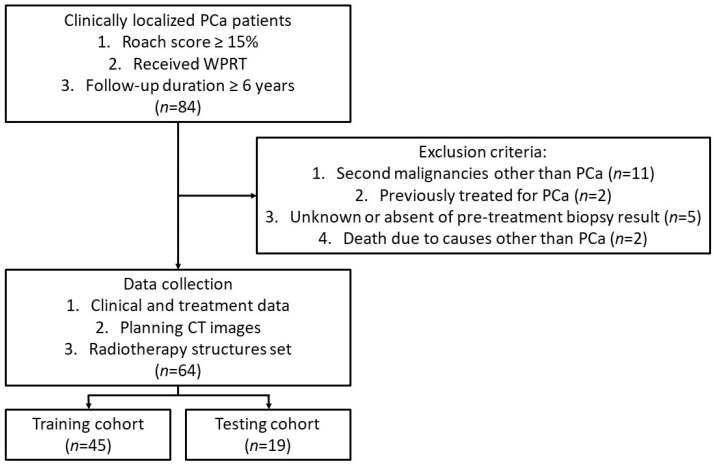
Patient selection procedures for radiomics model development. CT, computed tomography; PCa, prostate cancer; WPRT, whole pelvic radiotherapy.

**Figure 2 jpm-13-01643-f002:**
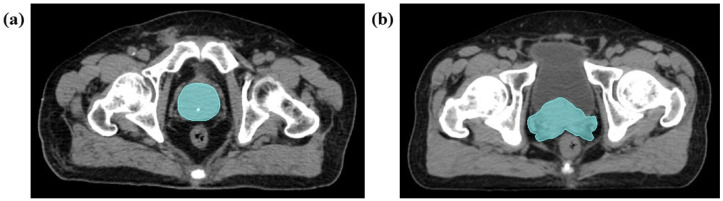
Axial planning computed tomography images with manually delineated CTV_prostate_ contours (green overlay). (**a**) CTV_prostate_ covering the entire prostate gland. (**b**) CTV_prostate_ including the entire prostate gland with a proximal two-thirds of the seminal vesicles.

**Figure 3 jpm-13-01643-f003:**
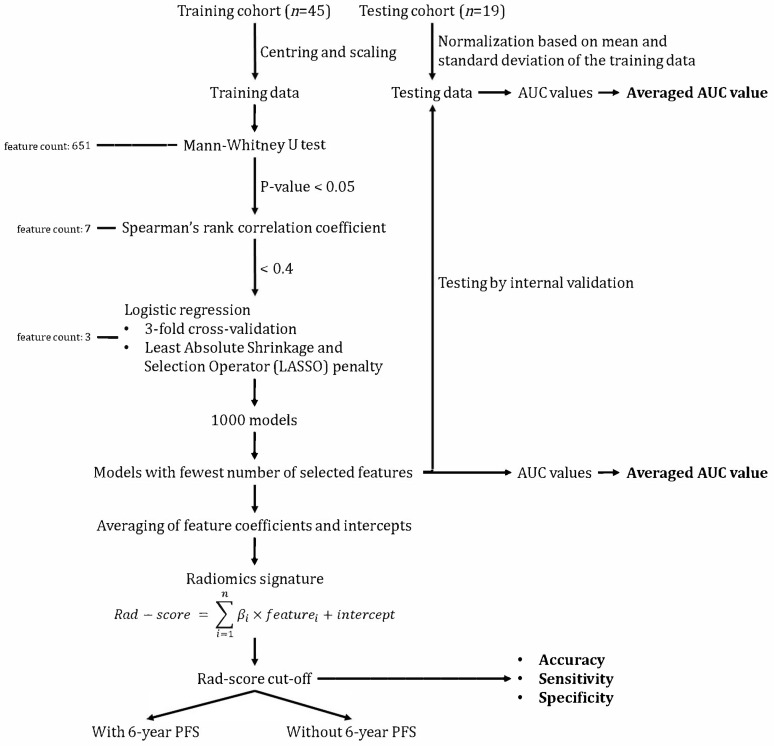
Feature selection, model development and statistical analysis workflow. AUC, area under receiver operating characteristic curve; PFS, progression-free survival.

**Figure 4 jpm-13-01643-f004:**
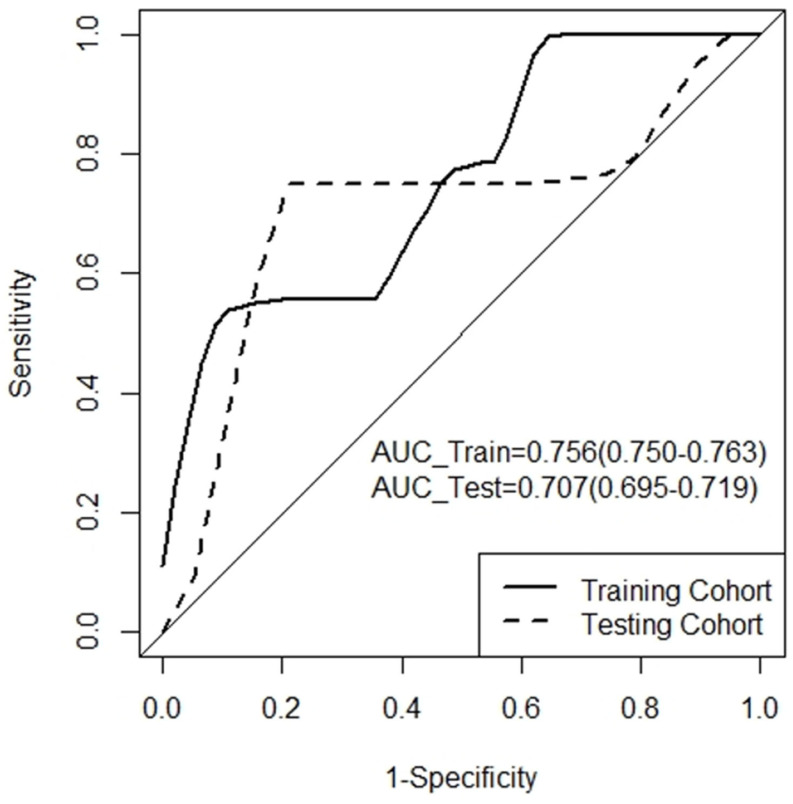
Receiver operating characteristic (ROC) curves of the developed model for training and testing cohorts. Figures in parentheses are 95% confidence intervals. AUC_Train, area under ROC curve (AUC) of training cohort; AUC_Test, AUC of testing cohort.

**Figure 5 jpm-13-01643-f005:**
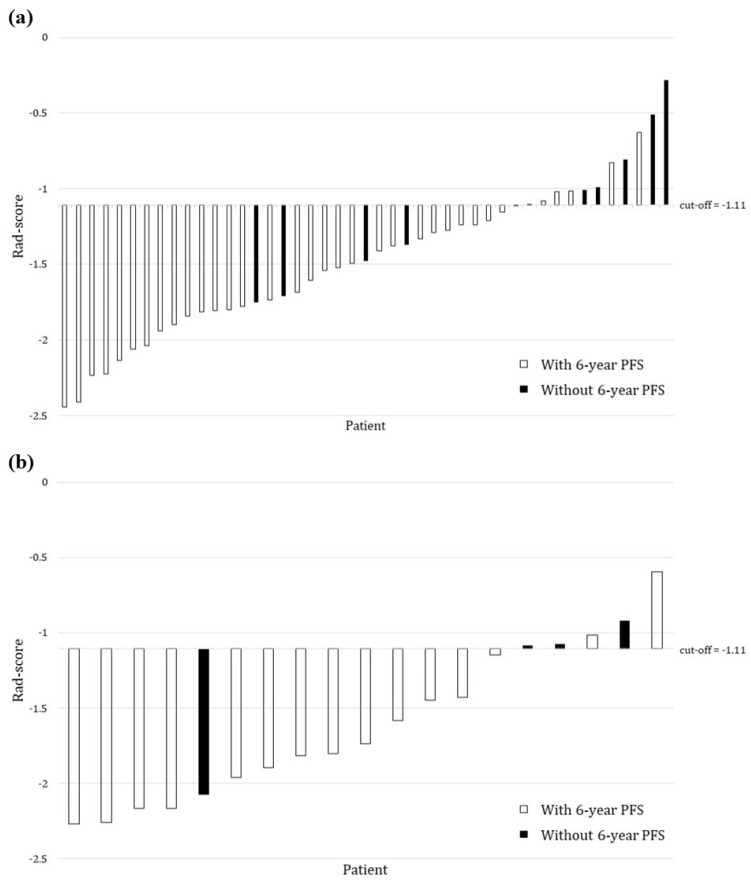
Rad-score charts for (**a**) training and (**b**) testing cohorts. PFS, progression-free survival.

**Table 1 jpm-13-01643-t001:** Patients’ clinicopathological characteristics.

Characteristic	All Included Patients (*n* = 64)	Training Cohort (*n* = 45)	Testing Cohort (*n* = 19)	*p*-Value
*Median age at start of radiotherapy (years)*	70	71	70	0.691 ^1^
*Histology*				
Adenocarcinoma	49 (76.6%)	31 (68.9%)	17 (89.5%)	0.160 ^2^
Acinar adenocarcinoma	14 (21.9%)	13 (28.9%)	2 (10.5%)	
Unknown	1 (1.6%)	1 (2.2%)	0 (0%)	
*Stage*				
T1	8 (12.5%)	6 (13.3%)	2 (10.5%)	0.189 ^2^
T2	20 (31.3%)	11 (24.4%)	9 (47.4%)	
T3	33 (51.6%)	25 (55.6%)	8 (42.1%)	
Unknown	3 (4.7%)	3 (6.7%)	0 (0%)	
*Pre-treatment PSA level (ng/mL)*				
<10	5 (7.8%)	2 (4.4%)	3 (15.8%)	0.052 ^2^
10–20	18 (28.1%)	16 (35.6%)	2 (10.5%)	
>20	41 (64.1%)	27 (60.0%)	14 (73.7%)	
*Pre-treatment GS*				
≤6	16 (25.0%)	10 (22.2%)	6 (31.6%)	0.476 ^2^
7	17 (26.6%)	11 (24.4%)	6 (31.6%)	
≥8	31 (48.4%)	24 (53.3%)	7 (36.8%)	
*Median pre-treatment Roach score*	32.8	33.6	27.9	0.130 ^1^
*Median CTV_prostate_ volume (mm^3^)*	39951.5	41695.0	38943.0	0.797 ^1^
*CTV_LN_ dose (Gy)*				
44	11 (17.2%)	7 (15.6%)	4 (21.1%)	0.719 ^3^
50	53 (82.8%)	38 (84.4%)	15 (78.9%)	
*CTV_prostate_ dose (Gy)*				
<76	12 (18.8%)	8 (17.8%)	4 (21.1%)	0.739 ^3^
≥76	52 (81.3%)	37 (82.2%)	15 (78.9%)	
*Treatment modality for PTV_LN_*				
3DCRT	10 (15.6%)	6 (13.3%)	4 (21.1%)	0.466 ^3^
VMAT	54 (84.4%)	39 (86.7%)	15 (78.9%)	
*Treatment modality for PTV_prostate_*				
IMRT	9 (14.1%)	5 (11.1%)	4 (21.1%)	0.432 ^3^
VMAT	55 (85.9%)	40 (88.9%)	15 (78.9%)	
*Patients received neoadjuvant ADT*	62 (96.9%)	44 (97.8%)	18 (94.7%)	0.509 ^3^
*Patients received adjuvant ADT*	52 (81.3%)	36 (80.0%)	16 (84.2%)	1.000 ^3^
*Median follow-up time (months)*	88.0	91.0	88.0	0.872 ^1^
*Median progression-free survival (months)*	81.5	81.0	85.0	0.659 ^1^
*Patients with six-year disease progression*	13 (20.3%)	9 (20.0%)	4 (21.1%)	1.000 ^3^
*Events*				
Biochemical recurrence	12	10	2	-
Local failure	3	2	1	
Regional failure	1	0	1	
Distant failure	5	5	0	

^1^ Mann–Whitney U test; ^2^ Chi-squared test; ^3^ Fisher’s exact test. 3DCRT, three-dimensional conformal radiotherapy; ADT, androgen deprivation therapy; CTV_LN_, clinical target volume for whole pelvic lymph nodes; CTV_prostate_, clinical target volume for prostate; GS, Gleason score; IMRT, intensity-modulated radiotherapy; PSA, prostate-specific antigen; PTV_LN_, planning target volume for whole pelvic lymph nodes; PTV_prostate_, planning target volume for prostate; VMAT, volumetric modulated arc therapy.

## Data Availability

The datasets used and/or analyzed in this study are available from the corresponding author on reasonable request.

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
