# Peer review of "Computed Tomography-Based Radiomics for Long-Term Prognostication of High-Risk Localized Prostate Cancer Patients Received Whole Pelvic Radiotherapy"

_jpm, 2023, doi:10.3390/jpm13121643_

Round 1
Reviewer 1 Report
Comments and Suggestions for Authors
The paper provides valuable insights into the investigation of planning computed tomography (pCT)-based radiomics for long-term prognostication of high-risk localized prostate cancer (PCa) patients undergoing whole pelvic radiotherapy (WPRT).
I have the following comments:
Abstract:
The methodology used in the study is not adequately described in the abstract. Although it acknowledges the utilization of best practice procedures for radiomics research, there is no further explanation regarding the specific actions, algorithms, or parameters employed in the development of the radiomics model.
Materials and Methods:
Medical Image Segmentation does not specify the specific tools, computer systems, or screens utilized in the segmentation process. The description mentions that the contours were initially contoured by one radiation oncologist and reviewed by another, with the involvement of an additional consultant radiation oncologist to adhere to best practice procedures. However, the specific details regarding the nature of segmentation, whether it was manual, semi-automatic, or involved a specific software tool, are not explicitly stated. Manual segmentation, for instance, might introduce inter-observer variability, while semi-automatic methods could have implications for consistency. The absence of information on the number of contours or slices segmented raises questions about the granularity of the segmentation process. How many slices were segmented? This could include intra- and inter-observer variability assessments, as well as any tools or guidelines utilized to ensure consistency across the segmented structures.
Feature Selection: It's important to note that the choice of a correlation coefficient threshold of ≥ 0.4 for feature removal is somewhat arbitrary and might benefit from additional justification or sensitivity analysis. Furthermore, providing insights into the potential impact of these feature selection choices on the model's interpretability and generalizability would enhance the transparency of the methodology.
Reviewer 2 Report
Comments and Suggestions for Authors
The Authors presented a very interesting study, however, it needs some improvement before it may be considered for publication in JPM.
1. Figure 1 is not graphically nice - the text should be nicely arranged in the middle of the boxes and the arrows should not cross the box lines as it looks so unprofessional.
2. The numbers of patients in Fig. 1 and Tab. 1 do not match. Please revise that.
3. The results of the study are not clearly presented. Maybe the Authors should add some graphs after Fig. 4 as there are no straightforward data that represents the final results of the study.
4. Discussion section should include more about the limitations of the study and about the potential inprecission of the model when considering other populations of patients.
Otherwise a nice paper!
Round 2
Reviewer 1 Report
Comments and Suggestions for Authors
The article was carefully revised and now it is in accordance for approval.
Reviewer 2 Report
Comments and Suggestions for Authors
The paper was improved and the Authors have addressed all my comments. I belive the paper may be accepted for publication now.